# Current Progress in *Sporothrix brasiliensis* Basic Aspects

**DOI:** 10.3390/jof9050533

**Published:** 2023-04-29

**Authors:** Manuela Gómez-Gaviria, José A. Martínez-Álvarez, Héctor M. Mora-Montes

**Affiliations:** Departamento de Biología, División de Ciencias Naturales y Exactas, Campus Guanajuato, Universidad de Guanajuato, Noria Alta s/n, col. Noria Alta, Guanajuato 36050, Mexico; manuela.gomezg8@gmail.com (M.G.-G.);

**Keywords:** animal sporotrichosis, diagnosis, immune response, pathogenic clade, treatment, zoonosis, immune sensing

## Abstract

Sporotrichosis is known as a subacute or chronic infection, which is caused by thermodimorphic fungi of the genus *Sporothrix*. It is a cosmopolitan infection, which is more prevalent in tropical and subtropical regions and can affect both humans and other mammals. The main etiological agents causing this disease are *Sporothrix schenckii*, *Sporothrix brasiliensis*, and *Sporothrix globosa*, which have been recognized as members of the *Sporothrix* pathogenic clade. Within this clade, *S. brasiliensis* is considered the most virulent species and represents an important pathogen due to its distribution and prevalence in different regions of South America, such as Brazil, Argentina, Chile, and Paraguay, and Central American countries, such as Panama. In Brazil, *S. brasiliensis* has been of great concern due to the number of zoonotic cases that have been reported over the years. In this paper, a detailed review of the current literature on this pathogen and its different aspects will be carried out, including its genome, pathogen-host interaction, resistance mechanisms to antifungal drugs, and the caused zoonosis. Furthermore, we provide the prediction of some putative virulence factors encoded by the genome of this fungal species.

## 1. Introduction

Mycoses, particularly those caused by dimorphic fungi, are considered a constant threat to public health [1]. Sporotrichosis is a mycosis that affects humans and other animals and is regarded as the most frequent subcutaneous fungal infection in Latin America [2]. Many types of clinical manifestations have been described for sporotrichosis; among them, the lymphocutaneous form is the most common in immunocompetent individuals, while the deep-seated infection is common in immunocompromised patients, increasing both morbidity and mortality rates [3,4]. For many years this disease was attributed to a single etiological agent, the species known as *Sporothrix schenckii*. However, phylogenetic studies revealed that this organism was a complex of cryptic species, four of which are of medical interest, *Sporothrix globosa*, *Sporothrix luriei*, *Sporothrix brasiliensis*, and *S. schenckii* [5,6]. Later, by molecular phylogenetic studies, these four species were grouped in the *Sporothrix* pathogenic clade [7]. The most studied species of this clade is *S. schenckii*. However, it was recently shown that the zoonotic outbreaks in Brazil are caused by *S. brasiliensis*, which has been recognized as the prevalent etiological agent of both feline and human sporotrichosis in the country. It is more thermotolerant than the other species of the genus [8,9,10,11,12,13]. *Sporothrix* species are characterized by showing a thermodimorphic phenotype; that is, in their saprophytic state, they can grow into a filamentous form, which is characterized by hyaline and septate hyphae that produce conidia, and during the parasitic phase when infecting the host tissues, fungal cells can be found as yeast-like cells (Figure 1) [14,15,16]. This dimorphism is known to be essential for fungal virulence in mammalian hosts [17].

Since 2015, the genome sequence of *S. brasiliensis* (strain 5110) has been available in the NCBI database (https://www.ncbi.nlm.nih.gov/data-hub/genome/GCF_000820605.1/, accessed on 2 March 2023) and the Genbank with access number AWTV00000000 [16]. The genome has a size of 33.2 Mbp, a GC percentage of 54.5%, and 9231 genes, of which 9091 code for proteins and 140 for functional RNAs [16]. When the *S. schenckii* and *S. brasiliensis* genomes were aligned in pairs, it was found that both species share a 97.5% identity [16]. The *S. brasiliensis* 5110 mitochondrial genome has 36 Kbb; however, it only covers 71–75% of the three previously reported *S. schenckii* genomes [16]. One of the differences between the *S. brasiliensis* and *S. schenckii* mitochondrial genomes is that the former contains homing endonucleases (HE’s), which encode parasitic group-I introns, providing a possible explanation for the larger size of its mitochondrial genome [16]. Fungal mitochondrial genomes are known to have constant genetic mobility, which can be attributed to HE activity. Introns encoding HE have been associated with changes in genomic size and mitochondrial parasitism, leading to genomic instability. These findings could explain some biological differences between *S. schenckii* and *S. brasiliensis* [16].

In recent years, the disease caused by *S. brasiliensis* has gained importance due to the reported epidemic outbreaks, especially those in Brazil [12]. The information available regarding the pathogen-host interaction and the *S. brasiliensis* virulence factors is until now limited, but in recent years this is gaining momentum. Although some fungal traits that affect pathogen-host interaction are known, such as the ability to generate melanin, susceptibility to antifungal drugs, adhesive properties, and cell wall composition and organization [18], more information is required to have a full picture of both basic and clinical aspects of *S. brasiliensis*. Therefore, in this review, we will focus on these characteristics and address aspects related to virulence factors, immune response, and antifungal drug resistance.

## 2. Virulence Factors

In different pathogenic fungi, various virulence factors have been described and are recognized as elements that the organism use during the host damage and whose absence can lead to a decrease in virulence [19]. Among the best-described virulence factors are adhesins, biofilm formation, hydrolytic enzymes, dimorphism, thermotolerance, immune evasion, melanin production, and proteins involved in cell wall synthesis [18,20]. The study of these virulence factors in *S. brasiliensis* is limited; however, to determine which of these factors could be found in this species, reference organisms such as *Candida albicans*, *Aspergillus fumigatus*, *Cryptococcus neoformans*, and the most phylogenetically similar species, *S. schenckii,* were selected and used to perform a Blastp analysis (https://blast.ncbi.nlm.nih.gov/Blast.cgi, accessed on 2 March 2023), to predict *S. brasiliensis* putative genes encoding virulence factors (Table 1).

The protein names are the accession codes of the sequences in the database of the National Center for Biotechnology Information (https://www.ncbi.nlm.nih.gov/, accessed on 2 March 2023). The putative protein encoded by the *Candida albicans*, *Aspergillus fumigatus*, and *Cryptococcus neoformans* genes were subjected to protein BLAST analysis (https://blast.ncbi.nlm.nih.gov/Blast.cgi?PROGRAM=blastp&PAGE_TYPE=BlastSearch&LINK_LOC=blasthome, accessed on 2 March 2023). The best hit was reported in the *Sporothrix schenckii* column, as previously reported [18]. The E-value and similarity columns refer to the comparison of amino acid sequences of the encoded proteins from *S. schenckii* with the putative *S. schenckii* ortholog.

Cell adhesion is considered an important characteristic of pathogens since it helps colonization and spread within the host [21]. In *Sporothrix*, the cell wall proteins have not been fully characterized; however, some reports have shown that this organism has some adhesins that can bind to the host extracellular matrix proteins, such as fibronectin, laminin, and type II collagen [22,23,24]. The adhesin most characterized in the genus *Sporothrix* is Gp70, which is found in the three most important species, *S. brasiliensis*, *S. schenckii,* and *S. globosa* (Table 2) [3,25,26]. This adhesin mediates *Sporothrix* adhesion to host tissues and basal lamina proteins [3]. The Gp70 expression has been related to the virulence of different *Sporothrix* species. Experiments carried out with high-virulence isolates of *S. brasiliensis* reported the limited presence of this adhesin; however, in low-virulent isolates of *S. brasiliensis* and *S. schenckii,* higher levels of this protein were observed [27]. This observation led to conclude that the adhesin properties may not be as relevant during the pathogenic process as its immunogenicity; i.e., the abundance of this protein on the cell surface could be detrimental to host colonization and invasion [27]. In silico comparative analyzes of the *S. brasiliensis* genome identified 54 proteins with adhesin function; however, these are currently annotated as hypothetical proteins or belong to a family of proteins with unknown functions [16]. It is necessary to carry out proteomic studies to validate the expression of these proteins and corroborate their cellular function [28]. Of the proteins found in this analysis, eight are exclusive of *S. brasiliensis*. The relevance of these proteins lies in the fact that they are not specific to human pathogenic fungi, but rather some of them show homology with proteins found in phytopathogens and entomopathogens [28]. These putative adhesins identified and predicted by ProFASTA and FungalRV were classified as proteins of cell wall maintenance, carbohydrate processing, proteolysis, nucleotidases, redox homeostasis, hydrolases, and ATPase-stabilizing factors [28]. Thus, if any of them have adhesion properties, they are likely to be moonlighting proteins [29]. It was recently determined that the *S. schenckii* chaperonin GroEl/Hsp60 and the uncharacterized protein Pap1 have adhesive properties to extracellular matrix proteins, such as laminin, elastin, fibrinogen, and fibronectin. The Pap1 can also bind to type I and type II collagen [23]. Through bioinformatic analysis, it was found that *S. brasiliensis* has putative functional orthologs for both proteins, SPBR_03666 and SPBR_07403, respectively (Table 2). Interestingly, Pap1 is found in *S. brasiliensis* and *S. schenckii* but not in *S. globosa*, which could provide a possible explanation for the low virulence associated with *S. globosa* strains [23].

Although some adhesins have been predicted in *S. brasiliensis*, the information is still scarce. Thus, bioinformatics analyses are important tools to elucidate the putative proteins related to cell adhesion in this species. The *S. brasiliensis* genome encodes several putative functional orthologs of adhesins from *C. albicans*, *A. fumigatus,* and *C. neoformans* (Table 1). Different adhesins have been described for *C. albicans*, and so far, the best-described adhesins in this species are agglutinin-like sequence (ALS) proteins, which belong to a gene family [33]. However, no functional orthologs for these ALS proteins were found within the *S. brasiliensis* genome (Table 1). No obvious functional orthologs for other *C. albicans* adhesins, such as Eap1, Hwp1, and Iff4, were found within the *S. brasiliensis* genome; however, for Ecm33, Int1, and Mp65 putative orthologs were identified (Table 1) [34]. The Als, Eap1, Hwp1, and Iff4 proteins play an important role in fungal adherence, biofilm formation, and antifungal resistance mechanisms [35,36]. The fact that *S. brasiliensis* does not have any obvious putative ortholog of these proteins could suggest that this species contains other adhesins that could be supplying these functions. Regarding *A. fumigatus* and *C. neoformans*, putative orthologs for the CalA, Scw11, Gel1, Gel2, and Mp98 genes were also found within the *S. brasiliensis* genome. For other adhesins such as RodA, RodB, AspF2, Mp1, AfCalAp, Cfl1, and Cpl1 [37], no obvious ortholog was found in *S. brasiliensis*, which could be related to the specific function that these proteins fulfill within these two fungal species.

Biofilms contribute to many aspects of the fungal life cycle. The formation of this begins with cell adhesion to biotic or abiotic surfaces, then fungal growth that goes in parallel with extracellular matrix production, and finally dispersion from biofilm to other surfaces [38]. *Sporothrix* spp. biofilms are structurally complex, showing a network of hyphae associated with the presence of an extracellular polymeric matrix and water channels. The hyphal network contributes to the structural maintenance of the biofilm architecture, and the water channel allows the transport of nutrients to cells found in the biofilm [39]. Compared to other microbial biofilms, biofilms formed by *Sporothrix* spp. have a slow growth rate [39]. Recent studies suggest that biofilm formation could be one of the most important factors involved in *Sporothrix* virulence [18]. The filamentous form of *S. brasiliensis* is capable of developing biofilms in vitro [39]. The *S. brasiliensis* yeast-like cell has a thick bilayered cell wall with high rhamnose and chitin content and long microfibrils, providing it with the adherence required to form biofilms [40]. The ability of *S. brasiliensis* and *S. schenckii* to form biofilms on pieces of cat’s claw has been evaluated. Both species showed different growth kinetics during the biofilm formation, and *S. schenckii* had higher metabolic activity during the first incubation time [41]. It was observed that *S. schenckii* and *S. brasiliensis* have different susceptibility profiles to antifungal drugs, the former being more susceptible in vitro than *S. brasiliensis*. These observations make it feasible to hypothesize that *Sporothrix*, especially *S. brasiliensis*, can form biofilms in the cat’s claws, representing a fundamental factor for the transmission of *Sporothrix* spp. [41]. Although biofilm formation could show structural variations among fungal species, it is known that in most of the medically relevant fungal species, this structure fulfills the same function, contributing to the virulence and survival of these organisms [42]. In *C. albicans,* biofilm formation is regulated through several genes such as *BCR1*, *BRG1*, *EFG1*, *HSP90*, *NDT80*, *ROB1*, and *CSR1*, while in *C. neoformans* this is controlled by *LAC1*, *URE1,* and *CAP59* [43,44,45,46]. According to our bioinformatic analysis carried out, these genes could be found within the *S. brasiliensis* genome (Table 1). As mentioned, both *S. schenckii* and *S. brasiliensis* can form biofilms [39], which is an indicator that these genes participate in this process. However, other genes found within the *Sporothrix* genome may also regulate this metabolic process.

Similar to other fungal pathogens, dimorphism is a highly important virulence factor for *Sporothrix* spp. [18]. Dimorphism gives it the ability to change from its saprophytic to the parasitic phase; that is, it changes from mycelial morphology to yeast morphology, adapting to the host milieu and spreading to different tissues [47]. For *S. brasiliensis,* it has been reported that thermal dimorphism is essential for infection establishment [16]. In other dimorphic fungi, such as *Histoplasma capsulatum* and *Talaromyces marneffei*, different genes have been described that are involved in dimorphism, including *RYP1*, *RYP2*, *RYP3*, *VEA1*, *PAKB,* and *HGRA* [48,49], these genes were also found in the *S. brasiliensis* genome [16]. In *S. schenckii,* it has been shown that calcium is related to dimorphism, which stimulates the fungal morphological transition to adapt to environmental changes [30,50]. In this species, a Ca^2+^/calmodulin (CaMK)-dependent protein, called Sscmk1, was found to be involved in the control of morphogenetic and proliferative processes [31,51]. This protein is also present in *S. brasiliensis* (SPBR_08459), and taking into account the percentage of similarity is 99% and an E-value of 0, this protein could fulfill the same function in this species (Table 2). Although some interesting characteristics of the dimorphism associated with *S. brasiliensis* are known, more studies are needed to elucidate the proteins related to dimorphism fully. Several transcriptional regulators of dimorphism have been studied in *C. albicans* and *C. neoformans*, including Cph1, Hgc1, Nrg1 and Tup1, Mob2, Cbk1, Tao3, and Sog2 [52,53]. According to our bioinformatic analysis, the *S. brasiliensis* genome contains genes encoding putative functional orthologs of these proteins (Table 1). The process that controls dimorphism in *S. brasiliensis* may likely be regulated by these genes and others that have not yet been identified in this organism.

Another phenotypic characteristic associated with virulence is thermotolerance. *Sporothrix* thermotolerance has long been considered an important virulence factor and is behind various clinical manifestations of sporotrichosis [54]. This virulence factor facilitates fungal cells to grow and colonize host cells at a controlled body temperature, which is generally higher than the optimum temperature, to maintain cell division in fungal pathogens [54]. For example, strains that grow at 35 °C but not at 37 °C are unlikely to be able to spread through the host lymphatic system [54]. Previous studies have reported that the thermotolerance of *S. schenckii* and *S. brasiliensis* is similar. In addition, it is mentioned that the thermotolerance of the latter does not seem to be related to the clinical forms of sporotrichosis, the spontaneous regression of the infection, or the state serology of the patients or sites of infection [54]. However, more recent studies have shown that severe forms of sporotrichosis are associated with host immunity, the size of the inoculum, thermotolerance, and virulence of the strain. *S. brasiliensis* is often associated with more severe cases of the disease and with atypical manifestations in both immunocompetent and immunocompromised patients [55]. Even though *S. brasiliensis* is more thermotolerant than the other *Sporothrix* species, it is necessary to gather more information to support the notion that this trait is behind fungal aggressiveness [27]. Currently, our knowledge about thermotolerance control in *S. brasiliensis* is limited. Through our Blastp analysis, it was possible to make an approximation of the possible genes that could be involved in thermotolerance in this species. This analysis suggests that the *S. brasiliensis* genome contains orthologs of genes involved in thermotolerance in the fungal species *C. albicans*, *A. fumigatus*, and *C. neoformans*, such as *HSP60*, *HSP104*, *SSA1*, *SSB1*, *CGRA*, *SCH9*, *HSF1*, *BIP/KAR2*, *SSC70*, *HSP88*, *BIP*, LHS1/ORP150, *HSP90* and *CCR4* (Table 1), which mostly code for heat shock proteins. In *S. schenckii*, the Hsp90 protein has already been studied, and it is known that together with Sscmk1, they contribute to thermotolerance in this species. Both proteins are also present in *S. brasiliensis* (Table 2), implying they could fulfill the same functions in this species.

The production of hydrolases, such as SAPs, lipases, phospholipases, and secreted hemolysins, is an important factor for fungal pathogenicity when interacting with the host since they contribute to the invasion of cells and tissues [56,57]. Some proteases, such as collagenase, gelatinase, and proteinase I and II, which hydrolyze human stratum corneum, collagen type I, and elastin, play an important role during the pathogen-host interaction, contributing to skin invasion and avoiding the anti-*Sporothrix* immune response [32,58,59]. Despite the importance of hydrolytic enzymes in pathogen-host interaction, little is known about these molecules in *S. brasiliensis*. Through our bioinformatic analysis, it was possible to identify putative *S. brasiliensis* orthologs of *C. albicans* and *A. fumigatus* genes that code for phospholipases, lipases, and proteinases (Table 1). According to the results of the Blast analysis, orthologs of Lip5-8, Sap1-8, Plb1-3, Pep1-2, Ap1, CtsD, and PlaA proteins are likely present in *S. brasiliensis* and used to carry out specific functions, as the invasion of host cells and evasion of the immune response, as has been reported for *C. albicans* and *A. fumigatus* [60,61]. However, it is necessary to carry out more experiments that help to elucidate the function of these genes in this species. The *S. schenckii* SPSK_06273 gene is upregulated in yeast cells, which could suggest that this is an important gene in protein degradation during host invasion [18]. Through the Blastp analysis, it was possible to establish that this gene is also found in *S. brasiliensis*, which could indicate that it may fulfill a similar function in this species (Table 1).

Immune evasion is a strategy that contributes to fungal virulence and involves several processes, such as biofilm formation, protease production, morphological changes, and protein synthesis [62]. For *S. brasiliensis*, proteins differentially expressed in the yeast form have been analyzed, focusing mainly on proteins involved in immune evasion [63]. The proteins identified have known roles in dimorphic transition, extracellular vesicle production, aerobic respiration, immunogenicity, and invasiveness. These proteins were described as aminopeptidase I, manganese superoxide dismutase, heat shock protein 70 kDa, glyceraldehyde-3-phosphate dehydrogenase (GAPDH), hydroxymethylglutaryl-CoA lyase, progesterone-binding protein, acetyl-CoA hydrolase, and rhamnolipid biosynthesis 3-oxoacyl-[acyl-carrier-protein] reductase [63]. Aminopeptidase I is a protein that weakens mammal defenses, and the differential expression of this protein in *S. brasiliensis* could be related to the high virulence of this species [3,63]. The 3-oxoacyl-[acyl-carrier-protein] reductase protein is involved in rhamnolipid biosynthesis, and purified rhamnolipids are known to interact with immune cells directly [63]. Although several proteins are known to be involved in *S. brasiliensis* immune evasion, more studies are needed to determine the identity of all the players in complex strategies. The Blastp analysis carried out here contributes to the elucidation of more proteins involved in this process, including putative orthologs of *C. albicans* Hgt1, Msb2, and Sit1, *A. fumigatus* Pksp/Alb1, and *C. neoformans* Rim 101 (Table 1).

Melanin is a fungal cell wall pigment that can mask the fungus against host immunity, contributing to pathogenesis and virulence [54,64]. The different morphologies of *Sporothrix* spp. can produce three different types of melanin: DHN-melanin, eumelanin, and pyomelanin. In addition to the already mentioned function, *Sporothrix* melanins also protect the fungus against different antifungal agents, such as amphotericin B, terbinafine, and nitrogen-derived oxidants [65]. In *S. brasiliensis,* the production of this pigment is associated with resistance to phagocytosis [54]. Under different experimental conditions, *S. brasiliensis* strains produce a higher amount of DHN-melanin than *S. schenckii* and tend to be more virulent for in vivo models of sporotrichosis [5,13,27,54,65,66]. Through the bioinformatics search, it was possible to elucidate the *S. brasiliensis* putative orthologs of the genes present in *A. fumigatus* that code for melanin synthesis, such as polyketide synthases and phenoloxidases FET3, TILA, and dihydrogeodin oxidase/laccase (Table 1).

The cell surface of pathogenic fungi plays an important role in the interaction with the host. The fungal cell wall is composed of glycoconjugates, structural polysaccharides, such as chitin and β-glucans, and cell wall glycoproteins [67]. Although the relevance of this structure for fungal biology is known, the information about the proteins that compose the *S. brasiliensis* cell wall is scarce. In silico analyses determined that *S. brasiliensis* may contain 117 different kinds of cell wall proteins, and analysis of its genome indicated that there is a unique *FKS* ortholog [16]. Moreover, our bioinformatics analysis identified putative orthologs to cell wall-encoding genes from *C. albicans* and *A. fumigatus* (Table 1). Some of these proteins have already been studied in *S. schenckii* [18]; thus, it is likely they may have similar roles during the biosynthesis of the cell wall. In the case of chitin synthesis, it is known that this wall component is regulated by multigenic families, which encode chitin synthase isoenzymes [68]. Within these families, mainly 7, Chs1 to Chs7, are recognized. According to our Blastp analysis carried out, *S. brasiliensis* may contain different members of the chitin synthase family (Table 1).

*S. brasiliensis* is the species with the highest virulence among the members of the *Sporothrix* pathogenic clade, followed by *S. schenckii* and *S. globosa* [5]. Some of the traits behind this observation have already been mentioned in this section, and these include better adhesion due to the characteristics conferred by the external structure of the cell wall and the production of melanin. Works based on the overexpression of enzymes involved in the metabolism of amino acids, lipids, and the positive regulation of glycolytic enzymes and trehalose synthase in *S. brasiliensis*, suggest that this species can make better use of nutrients during parasitism than *S. globosa*. In addition, it has better resistance to osmotic stress and cell wall remodeling, which are factors that positively influence the infectious process and that can contribute to *S. brasiliensis* virulence [47,69,70].

## 3. Zoonosis Caused by *Sporothrix brasiliensis*

Sporotrichosis is commonly known as “gardener’s disease.” However, the infection caused by *S. brasiliensis* goes beyond this typical definition [71]. *S. brasiliensis* is an emerging fungal pathogen associated with cat-to-human (zoonosis) and cat-to-cat/dog transmissions and has become a strong epidemic and epizootic infectious agent [12,71]. Frequently, sapronotic transmission, which is associated with the environment, is the most common form of human sporotrichosis; however, it has been reported that zoonotic infections have become more common due to the emergence of *S. brasiliensis* [10]. Unlike *S. schenckii*, where transmission occurs through traumatic inoculation of the fungus from contaminated plant material, *S. brasiliensis* is also transmitted through bites, scratches, or contact with exudates from skin lesions of an infected cat [72,73,74]. Moreover, atypical routes of infection, such as outdoor tattooing, have also been documented [75].

Sporotrichosis has acquired zoonotic potential, and concern about this infection is growing due to the different outbreaks in different geographical regions [76,77,78]. Recent studies indicate that South America is the subcontinent with the highest prevalence of animal sporotrichosis, comprising 81%, followed by Asia and Europe. In the case of North America and Africa, they report similar low proportions [79,80]. In regions of Brazil, such as Rio de Janeiro, Rio Grande do Sul, and São Paulo, feline sporotrichosis has increased, and due to its alarming proportions, it is considered an emerging zoonotic fungal disease [71]. From 1997 to 2001, 178 clinical cases were reported in Rio de Janeiro and neighboring municipalities [81]. In 1998, the first epidemic of sporotrichosis in Brazil (Rio de Janeiro) transmitted by cats was recorded, which was caused by *S. brasiliensis* [73,81]. From 2002 to 2004, the number of clinical cases tripled and reached 572 [81,82]. In the last 20 years, the infection caused by this pathogen in Rio de Janeiro spread out and affected 4700 cats and around 5400 humans [73,83,84]. Between 2010 and 2016, in Rio Grande do Sul, 374 cases of sporotrichosis were diagnosed in cats and 83 cases in humans [76]. In both Brazilian regions, the species that predominated was *S. brasiliensis* [73,76]. In urban areas with a high density of feline population, important cases of epizootics have been reported and associated with *S. brasiliensis* [10,11]. Although this infection is more common in Brazil, other countries, such as Argentina, Paraguay, Uruguay, Chile, and Panama, have reported cases caused by *S. brasiliensis* (Figure 2) [77,83,85,86,87]. In the United Kingdom (UK), a country where *S. brasiliensis* is considered a rare pathogen, a single case of infection was reported by a veterinarian. The veterinarian had direct contact one month earlier with a cat imported from Brazil, which presented ulcerated pustular skin lesions [88]. In addition, two other cases of *S. brasiliensis* transmitted by cats were recently reported in the same country. The patients with confirmed sporotrichosis had previously been scratched by their cat, which had been moved from Brazil to the UK. Considering that the UK is a country where this organism has never been reported, one of the possibilities is that the cat acquired this pathogen in Brazil and that an acquired immunodeficiency event from a virus such as FIV triggered a more invasive disease. The cat may have been asymptomatic upon arrival in this country, or the skin lesions were so small that during border control inspection, they could not be observed [89].

Zoonotic sporotrichosis has been documented in countries such as the United States, Malaysia, India, and Mexico; however, in these cases, the causative agent has been identified as *S. schenckii* [90,91,92]. Although most cases associated with this type of sporotrichosis are caused by *S. brasiliensis*, cases have been reported where the histopathological analysis determined that the infection may be caused by *S. schenckii* [88]. However, this could be related to an error in the identification of the causative agent since most of the cases reported for zoonosis are attributed to *S. brasiliensis*. Records of travel-associated *S. brasiliensis* infections are increasingly common, and this problem indicates the potential for this pathogen to spread to new areas, mainly through traveling with cats [92].

Based on information collected in recent years, it is clear that animal sporotrichosis caused by *S. brasiliensis* is more common in cats. The most frequent clinical signs in these animals are associated with the cutaneous form, which is characterized by erosions, ulcers, and fistulas, which are accompanied by bloody serous or purulent exudates [83,87]. These wounds are located mainly on the head, face, neck, and extremities [83,87,93]. The incidence in cats is related to several factors: the first is that cats are generally outdoors, in constant contact with nature. In addition, due to their biological characteristics, cats tend to scratch the vegetation, which facilitates fungal spread in the environment, and finally, due to their instinct, they tend to fight between males, which promotes direct contact of open wounds between them [87,94,95]. It is important to mention that although cats are the main source of infection, other mammals can also be affected (Figure 3) [74].

In the case of dogs, it has been shown that they are not an important source of human infections by *S. brasiliensis* [73]. Previous studies in Rio de Janeiro found that up to 84% of dogs with sporotrichosis had contact with cats before the manifestation of the infection [96]. From 1998 to 2014, 244 dogs (*Canis lupus familiaris*) were diagnosed with sporotrichosis in Rio de Janeiro; however, infection statistics in cats are much higher compared to these figures [10,73]. These cases were taken from a single institution, which does not reflect the real panorama of this disease in this Brazilian region. Dogs are reportedly not directly involved in the transmission of *Sporothrix* spp., which in most cases may be due to the low fungal load on their lesions [96]. Currently, the transmission of *Sporothrix* spp. by contact with infected dogs is anecdotic [97].

In a study where 103 clinical cases of animal sporotrichosis were analyzed, it was determined that of these, 92 were cats and 11 dogs. Both dogs and cats showed the cutaneous form of the disease, with respiratory and systemic complications. In dogs, the most frequent clinical form was fixed cutaneous infection in 54.5% of cases, followed by disseminated cutaneous and respiratory sporotrichosis in 18.2% of cases, and finally, lymphocutaneous infection in 9.2% of cases [97]. Dogs with the respiratory form of the disease showed sneezing and nasal discharge with no skin lesions, and one dog had a nasal plane deformity. This form of disease in dogs is the first reported in Brazil [97]. A general characteristic of the dogs that developed the disease is that they had free access to the outdoors and were in direct contact with other animals [97]. In other Brazilian states, such as Espírito Santo, Rio Grande do Norte, Mato Grosso, Minas Gerais, São Paulo, Paraná, and Santa Catarina, cases of canine sporotrichosis have also been described [94]. Several reports have proposed that other animals, such as armadillos, bats, rats, and squirrels, may also be potential carriers of *Sporothrix* propagules [10,98]. Rats (*Rattus norvegicus*) may play a role in the transmission of *S. brasiliensis* between cats [74,99]. This could potentiate outbreaks among humans, although it is necessary to have more evidence to confirm this affirmation. In Brazil and Argentina, cases of human sporotrichosis have been reported after rat bites [74,99]. Due to the constant interaction between rats and cats in urban environments, the appearance and spread of *S. brasiliensis* could be common [74]. In 2003, *S. brasiliensis* was isolated from a soil sample that was taken from an armadillo burrow (*Dasypus novemcinctus*) in Argentina [99]. Analysis of the stomach content of cats determined the presence of *S. brasiliensis*, and it was also found in cat feces collected from sand in São Paulo [71]. This feces can contaminate the soil, creating an environmental reservoir for *S. brasiliensis*, and thus developing a new possible source of contamination for both animals and humans [71]. Although sapronotic transmission of *S. brasiliensis* is poorly documented, recent studies have determined that this species can be found in the environment, contributing to this type of transmission [71].

Immunosuppression is a factor that influences the severity of the infection caused by *Sporothrix*. In the case of humans, disseminated sporotrichosis frequently occurs in immunocompromised persons; and this same phenomenon has recently been reported in feline sporotrichosis [100]. Feline immunodeficiency virus (FIV) or feline leukemia virus (FeLV) coinfections have been associated in some studies with a higher incidence of sporotrichosis or worse clinical outcomes [100]. Cats coinfected with these viruses show some differences in cytokine levels, showing higher levels of IL-10 and lower levels of IL-4, IL-12, and CD4^+^/CD8^+^, compared to other cats with sporotrichosis without coinfection with these viruses [100]. All the cats analyzed in this study had severe forms of the disease, were in poor health, and the main characteristic was that they were co-infected with FIV and FeLV [100,101]. It was further reported that cats with severe sporotrichosis but negative for FIV and FeLV could have other comorbidities and immunosuppressive conditions [100]. When histological analyzes of the samples were carried out, changes were found in the samples of skin lesions from cats with coinfection, suggesting a less efficient immune response compared to immunocompetent cats [101]. These changes could be associated with a reduced ability to eliminate fungal cells. Cats with viral co-infection had lower numbers of neutrophils and macrophages, a higher fungal burden, and an absence of well-formed granulomas [101]. These findings could be related to the fact that both FIV and FeLV are immunosuppressive viruses, which can cause neutropenia [101]. FeLV virus can cause changes in the phagocytic and chemotactic functions of neutrophils, resulting in poor healing of skin lesions [102]. When fewer neutrophils and a higher fungal load are present, the healing of skin lesions in cats with sporotrichosis may be impaired [101]. Some differences occur between young and adult cats. In cats older than five years, greater infiltration of lymphocytes and a greater frequency of well-formed granulomas were observed compared to younger cats. These results suggest that the immune response against *Sporothrix* might be less effective in young cats [101].

The population of stray cats in various countries of the world is large, and other morbidities apart from the mentioned viruses cannot be ruled out, such as tapeworms. Infection caused by tapeworms immunosuppresses cats and therefore increases their susceptibility to sporotrichosis [103]. Parasites, such as helminths, are inducers of immune anergy and defects in anti-inflammatory responses and are important agents that can drive a Th2-biased immune response [103].

Recent findings have shown that *S. brasiliensis* can maintain in environmental material for years [104]. Environmental studies were carried out in an area within the city of Rio de Janeiro, Brazil, where cases of human and feline sporotrichosis had occurred for 10 years. Wood samples were collected inside the house, and wood was found in the garden since it has been described that sporotrichosis in Brazil is associated with areas that have poor sanitary conditions [104]. The presence of *S. brasiliensis* in the analyzed house could be related to the surrounding forests that are being demolished, and it is possible that *S. brasiliensis* was already found in these and that a mammalian host could have been infected from the organic matter [104]. These data reported in Brazil are similar to those described in Argentina, where many cats with sporotrichosis were rescued from abandoned houses, which had wooden floors, and which could be a probable source of infection [86]. Previously, attempts had been made to isolate *S. brasiliensis* from organic matter; however, the isolation had not been successful, which could be related to the moderate growth of *Sporothrix* and the abundant growth of other saprophytic microorganisms [11,76].

To control the infection caused by this pathogen, cats must not be allowed to roam the streets freely, overcrowding must be avoided, and all sources of infection must be minimized. Cats showing symptoms of sporotrichosis should be tested with a variety of tests, beginning with retrovirus tests. Early identification of the FIV and FeLV viruses could bring benefits to patients with both diseases since antiviral therapy could avoid aggravating the problem.

## 4. Antifungal Resistance

*S. brasiliensis* and *S. schenckii* cell walls show differences, among them the amount of glycoprotein 70 kDa (gp70); *S. brasiliensis* has a thicker cell wall with higher rhamnose and chitin contents and longer wall microfibrils that contribute to the biofilm formation [27,40,74]. These characteristics could contribute to increased resistance to conventional antifungal drugs and virulence [8,27,40]. Few studies have focused on the mechanism of antifungal resistance in this fungal species [105]. Although these are not entirely clear, the development of resistance in *S. brasiliensis* could be related to several factors, such as the ability to produce melanin, genetic diversity, and mutations in cytochrome P450 [105].

*Sporothrix* species produce three types of melanin, DHN-melanin (I), L-DOPA-melanin (II), and pyomelanin (III). In the host, *Sporothrix* spp. can synthesize DHN and L-DOPA, which protect the fungus against the immune and antifungal response; and it is known that both can inhibit phagocytosis and macrophage death [106,107]. In *S. brasiliensis*, the biosynthesis of melanin is not fully elucidated, but it is known that its production is faster than in *S. schenckii* [105]. The production of the three types of melanin is related to lower susceptibility to the antifungal amphotericin B [65]. Melanin also protects *Sporothrix* species against the effect of terbinafine, an antifungal that can be used in some sporotrichosis cases [108].

In the case of genetic diversity, this has important consequences at the population, community, and ecosystem levels [109]. It is related to better adaptability under selection pressure in *Sporothrix* species [110,111]. In the *Sporothrix* clinical clade, high degrees of antifungal resistance have been reported, and phylogenetic analyzes of these species show that there is a selection process in the evolutionary past of *S. brasiliensis* and *S. globosa*, which are species that present less polymorphism compared to *S. schenckii* [112,113,114,115]. In vitro, antifungal susceptibility studies have revealed that *S. brasiliensis* and *S. schenckii* are more susceptible to antifungal drugs than *S. globosa* and *S. mexicana* [112]. In clinical cases associated with *S. brasiliensis*, it has been observed that they respond well to antifungal drugs with low MIC values [115,116]. However, considering selective pressure, several fungal isolates have shown polymorphisms in chromosome number and size, which is thought to play a role in the development of antifungal resistance genes [105]. Abnormal chromosome number could be another reason which offers extra copies of resistance genes that contribute to a resistant phenotype [117,118]. In southern Brazil, *S. brasiliensis* isolates are resistant to drugs such as ITZ, which could be associated with the genetic diversity of the species in this country [115]. Another source of antifungal resistance could be related to constant exposure to antifungals and host immunity, factors that can exert selective pressure on this organism [108,111].

Mutations in cytochrome P450 are another possible factor leading to antifungal resistance in *S. brasiliensis*. It is known that azoles fulfill the function of inhibiting cytochrome P450 monooxygenases, especially CYP51, which is directly involved in ergosterol synthesis [119]. In silico analyses focused on CYP51 revealed that mutations in the ITZ T230N binding site, which is the first-line antifungal to treat sporotrichosis, are associated with greater resistance to azoles, as has been shown in other fungi such as *Candida albicans* [105,119].

Although some antifungal resistance mechanisms are known, more studies are needed to elucidate these pathways fully and to establish differences that may occur between *Sporothrix* species.

## 5. Immune Response against *S. brasiliensis*

The host’s immune cells recognize pathogen-associated molecular patterns (PAMPs), a hallmark of the pathogenic invader, via the pattern-recognition receptors (PPRs). Once the host recognizes the invading agent, it triggers a series of reactions aimed at controlling or eliminating it.

Toll-like receptor-2 (TLR-2) is an important receptor involved in the recognition of pathogens, such as *Aspergillus*, *Pneumococcus*, and *Staphylococcus* [116]. *S. brasiliensis* is also recognized by TLR-2, and this receptor plays a key role in the protection against the fungus since the tlr-2^−/−^ mice bone marrow-derived macrophages (BMDMs) displayed limited phagocytosis capacity, indicating its importance in this process [120]. Nitric oxide (NO), a critical molecule to stimulate neutrophils and macrophages for killing and clearance of the pathogen, also shows decreased levels in tlr-2^−/−^ mice, emphasizing its importance in *S. brasiliensis* sensing [120]. The elimination of the invading agent depends, to a large extent, on the recruitment of immune cells through chemotaxis, which is favored by the release of cytokines and interleukins. TLR-2 loss caused a decrease in TNF-α, IL-6, and INF-γ levels during *S. brasiliensis* infection, which indicates that this receptor is involved in the production of proinflammatory cytokines [120]. Similar results were observed when human peripheral blood mononuclear cells (hPBMCs) were challenged with *S. brasiliensis* yeast-like cells, where TLR-2 participated in the production of TNF-α, IL-6, and IL-1β [121].

An immunoproteomic approach has shown that some *S. brasiliensis* peptides (ZR8 and ZR3) stimulate the production of INF-γ, IL-1β, and IL-17A in CD4^+^ T cells, demonstrating that these peptides may be candidates for a possible vaccine against sporotrichosis, especially that caused by the zoonotic route [122].

TLR-4 also participates in the immune recognition of human pathogenic fungi [123], and for *S. brasiliensis,* it is also an important PPR since the BMDMs from tl4 ^−/−^ mice are affected by its ability to engulf the fungus, and the NO levels are diminished [124], allowing fungal persistence at the infection site. The close relationship observed between high fungal burden and decreased levels of TNF-α, INF-γ, IL-6, and IL-10 in tlr-4^−/−^ mice demonstrate the importance of this receptor in the *S. brasiliensis* immune sensing [124]. Similarly, it has been demonstrated that the TLR-4 receptor participates in the recognition of *S. brasiliensis* by hPBMCs, where the levels of TNF-α, IL-6, IL-10, and IL-1β decreased when the receptor was blocked [121].

The *S. brasiliensis* cell wall peptide-rhamnomannan is an important glycoconjugate that stimulates cytokine production by hPBMCs. The induction of IFN-γ, IL-17, IL-22, IL-6, TNF-α and IL-1β is mediated by CR3, dectin-1 and TLR-4 receptors. The inflammasome, caspase-1, and the IL-1 receptor are also involved in the T-helper cytokines production [125]. Mannose receptor in hPBMCs is another important receptor to the proper production of the proinflammatory cytokines TNF-α and IL-6 during the interaction with the pathogen [121].

## 6. The Cat’s Immune Response against *S. brasiliensis*

The feline inflammatory immune response to combat an advanced sporotrichosis infection is inefficient, and the fungal load in the infection site is high. Immune cells, such as lymphocytes, macrophages, and neutrophils, have been found in deficient quantity when the mycosis is advanced, while in fungal granulomas that are not extensive, the quantity of immune cells is higher, in particular macrophages, which helps in controlling the infection [126,127]. Plasmocytes and mast cells have also been found in granuloma samples [101]. An important feature that may be related to feline susceptibility to *Sporothrix* spp., especially *S. brasiliensis*, is the minimal or lack of lymphoplasmacytic reaction that has been observed in these animals [128].

There are also other immune cells involved in feline sporotrichosis, such as monocytes, CD4^+^, CD8^low^ subset, and CD14^+^. In sporotrichosis cases, high levels of IL-4 have been found, which stimulate CD8 cell activation [129]. In the most severe cases of feline sporotrichosis, co-infection with FIV or FLV has been reported, which is known to cause feline immunosuppression with low CD4^+^/CD8^+^ ratio and decreased levels of IL-4 and IL-12 and high levels of the anti-inflammatory interleukin IL-10. This condition is similar to what has been observed in humans co-infected with *Sporothrix* and the Human Immunodeficiency Virus (HIV) [71,100,130]. Currently, the humoral immune response in cats infected with *S. brasiliensis* has not been studied, so there is little information available on this topic.

## 7. Concluding Remarks

Over the years, important advances have been made in the different biological aspects of the species belonging to the *Sporothrix* pathogenic clade. Although most of the attention has focused on *S. schenckii*, current knowledge of *S. brasiliensis* is increasing. As of 2015, the genome of this species is available in the NCBI database. Access to this information helps to continue elucidating important aspects of this species, such as proteins involved in different biological processes, possible virulence factors, and the interaction of this species with the host. In addition, the availability of this information may contribute to explaining the differences with other species of the *Sporothrix* pathogenic clade.

Although significant advances have been made in the study of the immune response of this species during sporotrichosis, much remains to be elucidated. To obtain more precise information about *S. brasiliensis*, it is necessary to know aspects such as virulence factors, which play an important role in host damage. Since these have not been experimentally elucidated, bioinformatics tools have become key to understanding these aspects. These tools make it possible to generate genetic predictions, which contribute to the detection of differences and similarities related to virulence, resistance to antifungal drugs, and relevant biological information when compared with other *Sporothrix* species. The possible virulence factors elucidated in this article could be useful in developing new diagnostic and treatment strategies and finding new therapeutic targets that help control sporotrichosis caused by *S. brasiliensis*. Finally, it is important to mention that more molecular, immunological, genetic, and epidemiological studies are needed to learn more about this species.

## Figures and Tables

**Figure 1 jof-09-00533-f001:**
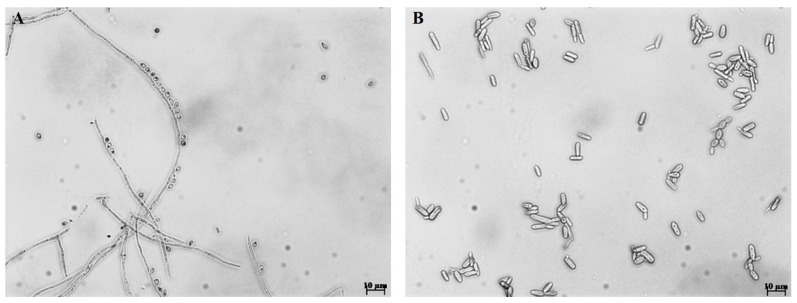
Microscopic morphology of *Sporothrix brasiliensis*. (**A**) Mycelium was grown in a liquid medium at 28 °C and pH 4.5 in the presence of conidia and branching septate hyphae. (**B**) Yeast-like cells growing at 37 °C and pH 7.8, with the typical elongated cigar shape. Scale bars 10 µm.

**Figure 2 jof-09-00533-f002:**
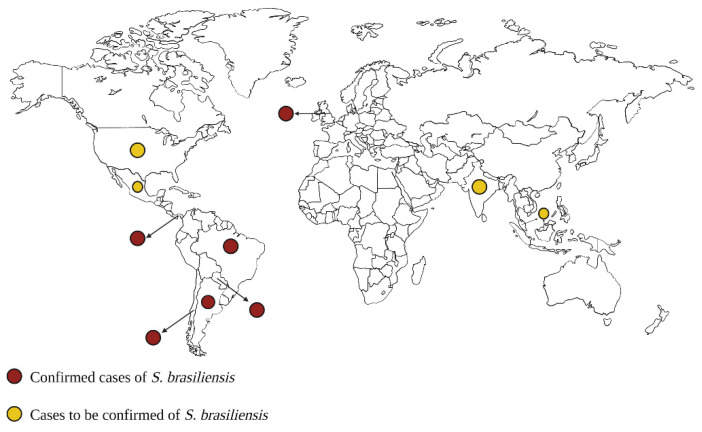
Geographic distribution of *Sporothrix brasiliensis* isolates. This species is distributed mainly in the American continent, in countries such as Brazil, Argentina, Paraguay, Panama, and Chile. Red dots indicate countries where *S. brasiliensis* has been isolated. The yellow dots are cases of sporotrichosis where the causative agent is yet to be confirmed, but the responsible species is likely *S. brasiliensis*.

**Figure 3 jof-09-00533-f003:**
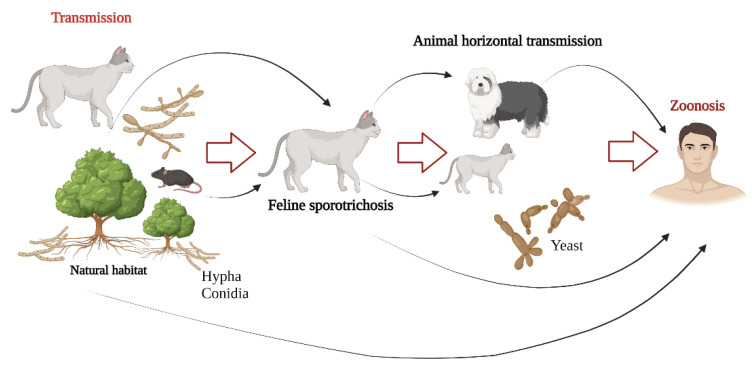
Transmission routes of *Sporothrix brasiliensis*. *Sporothrix brasiliensis* can be found in decomposing vegetal matter in the mycelial morphology (conidia and hyphae), and the cat, due to its biological and behavioral characteristics, can easily acquire it. This fungal species has been associated with epizootic outbreaks, mainly in dogs, cats, and rats. Feline sporotrichosis can be transmitted to humans (zoonosis) through scratches, bites, and contact with exudates from wounds, through which large amounts of yeast-like cells are inoculated into the host tissue.

**Table 1 jof-09-00533-t001:** Putative virulence factors encoded within the *Sporothrix brasiliensis* and *Sporothrix schenckii* genomes.

Virulence Factors	Organism	Protein	*Sporothrix schenckii*	*Sporothrix brasiliensis*	E-Value	Similarity (%)
Adhesins	*Candida albicans*	Als1 Als2	No found	No found	-	-
		Als5	No found	No found	-	-
		Eap1	No found	No found		
		Ecm33	SPSK_05317	SPBR_07321	0	99
		Hwp1	No found	No found	-	-
		Iff4	No found	No found	-	-
		Int1	SPSK_07346	SPBR_02186	0	99
		Mp65	SPSK_05120	SPBR_06946	0	99
	*Aspergillus fumigatus*	RodA RodB	No found	No found	-	-
		AspF2	No found	No found	-	-
		CalA	SPSK_05470	SPBR_07664	3e^−114^	76
		Scw11	SPSK_04001	SPBR_05627	0	97
		Gel1	SPSK_05276	SPBR_07245	0	99
		Gel2	SPSK_04169	SPBR_05893	0	99
		Mp1	No found	No found	-	-
		AfCalAp	No found	No found	-	-
	*Cryptococcus neoformans*	Cfl1	No found	No found	-	-
		Cpl1	No found	No found		
		Mp98	SPSK_03393	SPBR_00699	0	98
Biofilms	*C. albicans*	Bcr1	SPSK_01505	SPBR_04121	0	96
		Brg1	SPSK_05129	SPBR_07096	9e^−97^	99
		Efg1	SPSK_07078	SPBR_02388	0	98
		Hsp90	SPSK_08698	SPBR_08225	0	99
		Ndt80	SPSK_09140	SPBR_01757	0	99
		Rob1	SPSK_03010	SPBR_00136	0	97
		Csr1	SPSK_08605	SPBR_08179	0	97
	*C. neoformans*	Lac1	SPSK_03091	SPBR_00252	0	97
		Ure1	SPSK_00695	SPBR_06609	0	99
		Cap59	SPSK_09241	SPBR_08456	0	97
Hydrolytic enzymes	*C. albicans*	Lip5-8	SPSK_03375	SPBR_00635	0	97
		Sap1-8	SPSK_06273	SPBR_05010	0	97
		Plb1-3	SPSK_01063	SPBR_06837	0	97
	*A. fumigatus*	Pep1	SPSK_02149	SPBR_03405	0	98
		Pep2	SPSK_00526	SPBR_07236	0	100
		Ap1	SPSK_07865	SPBR_03022	1e^−114^	100
		CtsD	SPSK_01559	SPBR_03923	0	93
		PlaA	SPSK_02253	SPBR_03538	6e^−127^	86
Dimorphism	*C. albicans*	Cph1	SPSK_07311	SPBR_02347	0	100
		Hgc1	SPSK_05321	SPBR_07336	0	99
		Nrg1	SPSK_00519	SPBR_07150	0	96
		Tup1	SPSK_02314	SPBR_00318	0	93
	*C. neoformans*	Mob2	SPSK_01925	SPBR_03835	2e^−169^	100
		Cbk1	SPSK_06025	SPBR_04750	0	99
		Tao3	SPSK_02910	SPBR_00004	0	99
		Sog2	SPSK_03988	SPBR_05624	0	99
Thermotolerance	*C. albicans*	Hsp60	SPSK_01586	SPBR_03666	0	99
		Hsp104	SPSK_08586	SPBR_08170	0	98
		Ssa1	SPSK_08625	SPBR_01381	0	97
		Ssb1	SPSK_03121	SPBR_00285	0	87
	*A. fumigatus*	CgrA	SPSK_09995	SPBR_08746	5e^−88^	98
		Sch9	SPSK_10850	SPBR_08428	0	99
		Hsf1	SPSK_08498	SPBR_01315	0	97
		BiP/Kar2	SPSK_04019	SPBR_05650	0	98
		Ssc70	SPSK_03148	SPBR_00327	0	100
		Hsp88	SPSK_00430	SPBR_06639	0	99
		BiP	SPSK_06078	SPBR_04806	0	99
		Lhs1/Orp150	SPSK_02198	SPBR_03415	0	98
		Hsp90	SPSK_08698	SPBR_08225	0	99
	*C. neoformans*	Ccr4	SPSK_07136	SPBR_02436	0	96
Immune evasion	*C. albicans*	Hgt1	SPSK_06192	SPBR_04908	0	99
		Msb2	SPSK_07127	SPBR_02423	0	97
		Pra1	No found	No found	-	-
		Rbt5	No found	No found	-	-
		Sit1	SPSK_02970	SPBR_00416	0	98
	*A. fumigatus*	Hyp1/RodA	No found	No found	-	-
		Pksp/Alb1	SPSK_00653	SPBR_06313	0	97
	*C. neoformans*	Rim101	SPSK_07198	SPBR_02496	0	99
Melanin production	*A. fumigatus*	Fet3	SPSK_07279	SPBR_02574	0	98
		TilA	SPSK_04101	SPBR_05738	0	99
		Dihydrogeodin oxidase/laccase	SPSK_07219	SPBR_02517	0	98
Cell wall synthesis	*C. albicans*	Fks1	SPSK_01365	SPBR_04029	0	98
		Dpm3	SPSK_02816	SPBR_04500	7e^−62^	98
		Pmt2	SPSK_08548	SPBR_01344	0	97
	*A. fumigatus*	ChsG	SPSK_06989	SPBR_02297	0	99
		ChsA	SPSK_08492	SPBR_08106	0	99
		ChsF	SPSK_04841	SPBR_06424	0	99
		Dpm2	SPSK_08145	SPBR_03330	6e^−54^	100
		Pmt1	SPSK_05892	SPBR_04624	0	98
		Pmt4	SPSK_08628	SPBR_08186	0	99
		Kre2/Mnt1	SPSK_09069	SPBR_08384	0	98
		Ktr4	SPSK_05332	SPBR_07360	0	100
		Och1	SPSK_03245	SPBR_00480	0	97
		Mnn9	SPSK_09403	SPBR_08521	0	100

**Table 2 jof-09-00533-t002:** Prediction of putative virulence factors in *S. brasiliensis* based on *S. schenckii* known virulence factors.

*S. schenckii* Virulence Factors	Protein	*S. brasiliensis*	E-Value	Similarity (%)	Function	References
Adhesins	Gp70	SPBR_08225	0	99	Adhesin with the binding capacity to fibronectin, laminin, and type II collagen	[25,27]
	Hsp60	SPBR_03666	0	99	Adhesin that binds to laminin, elastin, fibrinogen, and fibronectin	[23]
	Pap1	SPBR_07403	1 × 10^−106^	79	Adhesin that binds to laminin, elastin, fibrinogen, fibronectin, and types I and II collagen	[23]
Thermotolerance	Hsp90	SPBR_08225	0	99	Response to heat shock and proteotoxic stress	[30]
	Sscmk1	SPBR_08459	0	99	Morphological switching and thermotolerance	[31]
Proteases	Proteinase I	SPBR_05754	0	99	Proteases that hydrolyze human stratum corneum, type I collagen, and elastin. They play an important role in pathogen-host interaction.	[32]
	Proteinase II	SPBR_00540	0	96		

## Data Availability

Not applicable.

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
