# Peer review of "Current Progress in Sporothrix brasiliensis Basic Aspects"

_jof, 2023, doi:10.3390/jof9050533_

Round 1

Reviewer 1 Report

Dear authors

Specific comments

Page 1, line 40: Please replace "complex" with "genus".

Page 6, line 169: Please replace "sporotrichosis" with "Sporothrix spp."

Page 9, line 312: Please delete "or animal".

Page 9, line 318: Page 9, line 318: Please write "São Paulo" instead of "San Pablo".

Page 9, lines 323-325: In the period of 1998–2020, 5,393 human patients were diagnosed with sporotrichosis at INI/FIOCRUZ, Rio de Janeiro.

Gomes Izoton CF, de Brito Sousa AX, Valete CM, Schubach AO, Procópio-Azevedo AC, Zancopé-Oliveira RM, de Macedo PM, Gutierrez-Galhardo MC, Castro-Alves J, Almeida-Paes R, Martins ACDC, Freitas DFS. Sporotrichosis in the nasal mucosa: A single-center retrospective study of 37 cases from 1998 to 2020. PLoS Negl Trop Dis. 2023 Mar 27;17(3):e0011212. doi: 10.1371/journal.pntd.0011212. Epub ahead of print. PMID: 36972287.

The same update should be performed for the number of cases of feline sporotrichosis from Rio de Janeiro.

Page 9, lines 324-6: Please replace "felines" with "cats".

Page 10, line 350: Please replace "felines" with "cats".

Page 10, line 359: Would the term "reservoir" really be the most appropriate for the cat? I think "main source of infection" better reflects this situation. Another important thing is that only the sick cat is epidemiologically important for the transmission of S. brasiliensis.

Pge 10, lines 361-2: I suggest inserting manuscripts on canine sporotrichosis caused by S. brasiliensis. Similar to feline cases of sporotrichosis in Brazil, canine cases are almost exclusively caused by S. brasiliensis.

Page 11, line 368: “.…involved in transmission of Sporothrix spp....”

Page 11, line 369: “...due to the LOW fungal load...”

Page 11, line 369: “...the transmission of Sporothrix spp...”

Page 12, lines 402-3: So far, the presence of S. brasiliensis in the environment is poorly documented. I suggest making this clear in the text.

Page 13, line 469: I suggest replacing "gold standard" with "reference method".

Page 13, lines 479-83: These stains are generally used in histopathological analysis. Make this clear in the text.

Page 14, line 515: Please replace "pathologies" with "diseases".

Page 15, line 584: Please replace "zoonotic" with "animal".

Page 15, lines 584-90: In fact, few therapeutic studies have included a reasonable number of cats treated with itraconazole. The literature reports varied clinical cure rates for itraconazole in the treatment of feline sporotrichosis.

Page 15, lines 585-7: Souza and colleagues (2018) (reference 145) described that there was a significant decrease in fungal burden after treatment with itraconazole. Please correct the text.

Page 15, line 599: According to Reis and colleagues (2016) (reference 146) the treatment used in this study was the combination of KI capsule with itraconazole. KI capsule as monotherapy can also be an effective alternative, but clinical cure rates are higher when associated with itraconazole.

Author Response

We thank the Reviewer for his/her constructive comments that have improved our manuscript.

Dear authors

Specific comments

Page 1, line 40: Please replace "complex" with "genus".

Reply: done

Page 6, line 169: Please replace "sporotrichosis" with "Sporothrix spp."

Reply: done

Page 9, line 312: Please delete "or animal".

Reply: done

Page 9, line 318: Page 9, line 318: Please write "São Paulo" instead of "San Pablo".

Reply: done

Page 9, lines 323-325: In the period of 1998–2020, 5,393 human patients were diagnosed with sporotrichosis at INI/FIOCRUZ, Rio de Janeiro.

Gomes Izoton CF, de Brito Sousa AX, Valete CM, Schubach AO, Procópio-Azevedo AC, Zancopé-Oliveira RM, de Macedo PM, Gutierrez-Galhardo MC, Castro-Alves J, Almeida-Paes R, Martins ACDC, Freitas DFS. Sporotrichosis in the nasal mucosa: A single-center retrospective study of 37 cases from 1998 to 2020. PLoS Negl Trop Dis. 2023 Mar 27;17(3):e0011212. doi: 10.1371/journal.pntd.0011212. Epub ahead of print. PMID: 36972287.

The same update should be performed for the number of cases of feline sporotrichosis from Rio de Janeiro.

Reply: the change was done

Page 9, lines 324-6: Please replace "felines" with "cats".

Reply: done

Page 10, line 350: Please replace "felines" with "cats".

Reply: done

Page 10, line 359: Would the term "reservoir" really be the most appropriate for the cat? I think "main source of infection" better reflects this situation. Another important thing is that only the sick cat is epidemiologically important for the transmission of S. brasiliensis.

Reply: We agree with the Reviewer’s comment and the text has been amended accordingly.

Pge 10, lines 361-2: I suggest inserting manuscripts on canine sporotrichosis caused by S. brasiliensis. Similar to feline cases of sporotrichosis in Brazil, canine cases are almost exclusively caused by S. brasiliensis.

Reply: We agree with the Reviewer’s comment and this subject was included in the original version of our manuscript (now in lines 378-387 of the revised manuscript).

Page 11, line 368: “.…involved in transmission of Sporothrix spp....”

Reply: the change was done

Page 11, line 369: “...due to the LOW fungal load...”

Reply: the change was done

Page 11, line 369: “...the transmission of Sporothrix spp...”

Reply: the change was done

Page 12, lines 402-3: So far, the presence of S. brasiliensis in the environment is poorly documented. I suggest making this clear in the text.

Reply: the change was done

Page 13, line 469: I suggest replacing "gold standard" with "reference method".

Reply: done

Page 13, lines 479-83: These stains are generally used in histopathological analysis. Make this clear in the text.

Reply: done

Page 14, line 515: Please replace "pathologies" with "diseases".

Reply: done

Page 15, line 584: Please replace "zoonotic" with "animal".

Reply: done

Page 15, lines 584-90: In fact, few therapeutic studies have included a reasonable number of cats treated with itraconazole. The literature reports varied clinical cure rates for itraconazole in the treatment of feline sporotrichosis.

Reply: We agree with the Reviewer’s comment and the has been amended to clarify this point.

Page 15, lines 585-7: Souza and colleagues (2018) (reference 145) described that there was a significant decrease in fungal burden after treatment with itraconazole. Please correct the text.

Reply: the change was done

Page 15, line 599: According to Reis and colleagues (2016) (reference 146) the treatment used in this study was the combination of KI capsule with itraconazole. KI capsule as monotherapy can also be an effective alternative, but clinical cure rates are higher when associated with itraconazole.

Reply: We agree with the Reviewer’s comment and the has been amended to clarify this point.

Reviewer 2 Report

Comments to authors:

The review is clear and provides information about basic and clinical aspects of Sporotrhix brasiliensis.

This review should be useful not only for clinical laboratory practitioners, but also for clinical physicians.

Author Response

There are no comments to be addressed, thanks for your revision though.

Reviewer 3 Report

The manuscript by Manuela Gómez-Gaviria, José A. Martínez-Álvarez, and Héctor M. Mora-Montes aims to discuss and review progresses in the knowledge of Sporothrix brasiliensis and its effect on the clinical and diagnostic aspects. Sincerely, the manuscript has ups and downs, with a clear emphasis on the virulence aspect of this species, very well conducted and written. The contribution the authors try to bring reviewing the diagnosis and treatment of the disease is unnecessary and superficial, especially the treatment. In both sections, there are fragile usage of references and erroneous concepts, what compromise the valuable contributions of the other sections. Authors should focus on their strengths and adjust other relevant sections.

Major initial issues to be corrected:

Please, in some parts of the text, consider changing ‘spp’ to ‘spp.’. This represents an abbreviation of species, so, it has the ‘.’. This appears mostly in the beginning, and once more in the section ‘The cat´s immune response’.

 Title - The title should be changed to something similar to "Current progress in Sporothrix brasiliensis virulence aspects and host immune response"

Abstract

Line 15 – Panama is Central America, please, consider rewriting the sentence.

Introduction

In lines 28-30 – “Many types of clinical manifestations have been described for sporotrichosis, among them, the lymphocutaneous form is the most common in immunocompetent individuals; whilst the deep-seated infection is common in immunocompetent patients, increasing both morbidity and mortality rates [3,4]. The second “immunocompetent” is probably “immunocompromised/ immunosuppressed”.

In lines 40-41 – The affirmative “The pathogenic clade members are characterized by showing a thermodimorphic phenotype…” Aren’t the species of the environmental clade also thermodimorphic? Please, consider rewriting.

Check Figure 1 – Panel A should bring a better picture of the classical conidiophore disposition with the conidia organized in a flower-like aspect.

Figure 1 legend – italicize Sporothrix brasiliensis.

Virulence

Lines 119-120 – Consider improving the English correctness of the sentence: “Thus, if any of these has adhesion properties, it is likely that are moonlighting proteins.”. Maybe “Thus, if any of them have adhesion properties, they are likely to be moonlighting proteins.”

Zoonosis

Line 302 - Italicize Sporothrix brasiliensis.

Line 318 – Change ‘San Pablo’ to ‘São Paulo’.

Line 331 – In addition to the reference 89, there is a quite recent publication (https://doi.org/10.1016/j.mmcr.2022.12.004) with the report of three human zoonotic cases in the UK (one of them was reported in reference 89). Please, update the considerations of zoonotic sporotrichosis in the UK based on this new case series.

Lines 337-339 – Please, double check if the following sentence is clear, as it may sound a bit controversial: “Although most cases associated with this type of sporotrichosis are caused by S. brasiliensis, it is possible that misidentification wrongly assigned S. schenckii as the causative agent instead of S. brasiliensis.”. You mean that some of the cat-transmitted cases attributed to S. schenckii may have been actually caused by S. brasiliensis, due to a molecular misidentification? If so, the number of S. brasiliensis-cases would be even higher, as well as the involved countries.

Line 340 – Italicize the first S. brasiliensis.

Line 345 – Consider changing “and” to “in”, in “…American continent, and countries such as Brazil…”.

Considering figure 2 and its associated text, do you really think that S. brasiliensis is the likely species for the human zoonotic cases in Malaysia? When the molecular investigation was performed in that country, S. schenckii was always the identified species.

Figure 3 – Consider keeping just one dog, since two may bring an idea that dogs are more affected than cats, or that they have a huge role in the transmission. Also, the rat should be positioned before in the transmission cycle. If it is present, it should be one of the sources of acquisition to cats, together with the environment and other cats. The dog (keep just one), could be better positioned. Remember to illustrate the paths: environment to cat; environment to human; cat to cat; cat to dog; cat to human; environment to rat, rat to cat.

Lines 380-382 – Consider replacing ‘at’ before the percentages by ‘in’.

Line 383 – Check the sentence “Dogs with the respiratory form of the disease had to sneeze, and nasal discharge with no skin lesions, and one dog had a nasal plane deformity.”. ‘Had to sneeze’ seems incorrect… had sneezing, presented sneezing?

The diagnosis (THIS TOPIC SHOULD BE EXCLUDED)

Authors should focus in other topics. Nevertheless, in the next lines you can find some observations concerning diagnosis.

Lines 460-461 – The time for a diagnosis is higher than 5-8 days. This is just the time for the hyphae to appear… then, we have the days for the thermoconversion… the real total time is usually 14-21 days. Low sensitivity? Culture has excellent sensitivity.

Lines 467-469 – Check the ‘recently’. It´s hard to understand the sentence… sporotrichin was used decades ago, for example.

Lines 478-481 – The sentences “To better visualize fungal structures, Schiff and Gomori periodic acid methenamine silver stains can also be used, however, for human samples, this strategy has low sensitivity [77,113]. In the case of samples from infected cats, this technique turns out to be successful, because the fungal load is higher and the sensitivity of the test reaches 87% [114].” are not correctly written and bring erroneous results from reference 114. A correct form would be: “To better visualize fungal structures, periodic acid-Schiff and Gomori methenamine silver stains can be used in the histopathology. Giemsa technique or the Quick Panoptic method can be used for direct examination.  However, for human samples, this strategy has low sensitivity [77,113]. In the case of samples from infected cats, Pereira et al. successfully used the Quick Panoptic method with 79% sensitivity, what can be attributed to the high fungal load [114].”

484-496 – If it is needed to grow the fungus in culture, then, the molecular methods cannot optimize the time for diagnosis. If a method can be used in clinical samples, then it works as a rapid test. This paragraph should be organized to keep the adequate ideas. It brings concepts of a regular PCR mixed with restriction fragment techniques.

508-510 – Ineffective? Why? It´s exactly the opposite: serology helps mostly when there is extracutaneous sporotrichosis.

518-534 – It seems there is a mixture of the results from the ELISA diagnosis from Fundação Oswaldo Cruz (2007 – ref. 126) and those from the State University of Rio de Janeiro (2005 – ref. 128)

Lines 558-561 – The authors should add that a rapid diagnosis would benefit the basic health units, not only hospitals. If we need a solution for the public health, units closer to the population essentially need to rapid diagnose cases.

Treatment (THIS TOPIC SHOULD BE EXCLUDED)

Authors should focus in other topics. Nevertheless, in the next lines you can find some observations concerning treatment (for a clear understanding of my concerns to it).

The mixture of human and animal treatments is not appropriate.

Lines 564-568 – Please, update/adequate the information on the dosage of itraconazole since the dosage of 100 mg/day can be considered from the initial of treatment for localized sporotrichosis. There are articles, mostly from Rio de Janeiro, showing a complete response to treatment with 100 mg/day in localized forms of sporotrichosis (fixed and lymphocutaneous). Recently, Orofino-Costa et al., published Recommendations on sporotrichosis, from the Brazilian Society of Dermatology, pretty much based on the experience with cases caused by S. brasiliensis in that region (doi: 10.1016/j.abd.2022.07.001).  Your references 83, 106, 109, and 123, as well as the article doi: 10.1093/cid/cir245, are examples of the success of this dosage.

Lines 572-574 – The same update is needed for terbinafine, based on some of the same articles, added by doi: 10.1007/s11046-010-9380-8 and doi: 10.1111/j.1468-3083.2009.03306.x. Dosages from 250 mg/day proved efficacy in treating localized forms.

Lines 574-577 – Reference 141 seems inadequate to the topic ‘treatment’, considering it is a case report from a probable zoonotic infection due to a cockatiel. It should be a good reference for the zoonotic transmission part, but not here. From the same 1st author, Fichman, there are two other articles exploring cryosurgery both in general cases (doi: 10.1111/bjd.17532), with the elderly involved, and specifically for pregnant women (doi: 10.1371/journal.pntd.0006434). Maybe you were trying to cite one of these references. Another issue, is concerning reference 142, which is cited, but could be better explored, mentioning the electrosurgery as one of the adjuvant therapies for sporotrichosis.

Line 582 – ‘cost-effectiveness’ is not true anymore for itraconazole. It is reported to be the cheapest option, together with potassium iodide. Cheaper than terbinafine, for example. This is still an issue for newer azoles, like posaconazole.

Line 582 – restricted ‘in’ patients, not ‘to’ patients. The meaning turns opposite if ‘to’ is used, bringing the idea that azoles are adequate and exclusive for the patients with liver disease, heart failure, older adults, and pregnant women.

Line 591 – saturated solution (not solutions).

Lines 598-599 – Consider checking the observation about the potassium iodide in capsules for cats to bring a better information, because its success is when used together with itraconazole, not alone.

Lines 600-607 – Consider joining information in this paragraph with those from lines 574-577. They all mention adjuvant, non-pharmacological therapies.

Line 600 – Photodynamic therapy (not therapies).

Line 601 – procedure (not procedures). And, concerning its complexity, it is almost the same as what it happens with electrosurgery. Both require some expertise from the medical doctor, but since the professional is trained and experienced, they are safe, effective and relatively simple to perform.

In the last paragraph, the authors put together some perspective in the treatment of sporotrichosis, but they should emphasize that as a perspective, not as a treatment.

Antifungal resistance is number 6

Line 612 – I suggest attenuating the sentence to: ‘These particular characteristics of S. brasiliensis might contribute to the increase in resistance to conventional antifungal drugs, and to virulence [29,42,111].”

Immune response… is number 7

Line 673 – participates or participated?

Line 680 – change from “…also is an important” to “… it is also an important”.

Line 683-684 – demonstrates (it refers to close relationship).

Line 688 – rhamnomannan.

Lines 689-690 – the induction… is mediated.

The cat´s… is number 8

Concluding remarks … 9

Line 723 – contribute to explain.

References

Some references have scientific words to be italicized.

Reference 109 – line 1023 – ‘Brazilian’ instead of ‘Bbrazilian’.

References 122 and 127 are the same.

Reference 164 – line 1175-1176 – check the format.

Reference 173 – line 1200 – ‘Brazilian’ instead of ‘brazilian’.

Author Response

The manuscript by Manuela Gómez-Gaviria, José A. Martínez-Álvarez, and Héctor M. Mora-Montes aims to discuss and review progresses in the knowledge of Sporothrix brasiliensis and its effect on the clinical and diagnostic aspects. Sincerely, the manuscript has ups and downs, with a clear emphasis on the virulence aspect of this species, very well conducted and written. The contribution the authors try to bring reviewing the diagnosis and treatment of the disease is unnecessary and superficial, especially the treatment. In both sections, there are fragile usage of references and erroneous concepts, what compromise the valuable contributions of the other sections. Authors should focus on their strengths and adjust other relevant sections.
Reply: We thank the Reviewer’s critical comments, which we no doubt have improved the quality of our manuscript. Please see reply to your comments below.
Major initial issues to be corrected:
Please, in some parts of the text, consider changing ‘spp’ to ‘spp.’. This represents an abbreviation of species, so, it has the ‘.’. This appears mostly in the beginning, and once more in the section ‘The cat´s immune response’.
Reply: done.
Title - The title should be changed to something similar to "Current progress in Sporothrix brasiliensis virulence aspects and host immune response"
Reply: We thank you for this suggestion. At the beginning of your comments, you mentioned that some areas of the manuscript, mainly those based on clinical aspects of the disease should be modified to be at the levels of other sections of the manuscript (those addressing basic aspects). Later, you suggested removing the section dealing with clinical aspects. This would place the title in the same line as this suggestion. However, we prefer to adhere to your first recommendation and to improve the quality of sections dealing with clinical aspects. Therefore, we respectfully disagree with this title modification and removal of sections. We hope that you find the changes in those sections with improved quality.
Abstract
Line 15 – Panama is Central America, please, consider rewriting the sentence.
Reply: done.
Introduction
In lines 28-30 – “Many types of clinical manifestations have been described for sporotrichosis, among them, the lymphocutaneous form is the most common in immunocompetent individuals; whilst the deep-seated infection is common in
immunocompetent patients, increasing both morbidity and mortality rates [3,4]. The second “immunocompetent” is probably “immunocompromised/ immunosuppressed”.
Reply: The Reviewer is correct, and the text has been amended.
In lines 40-41 – The affirmative “The pathogenic clade members are characterized by showing a thermodimorphic phenotype…” Aren’t the species of the environmental clade also thermodimorphic? Please, consider rewriting.
Reply: The Reviewer is correct, and the text has been amended.
Check Figure 1 – Panel A should bring a better picture of the classical conidiophore disposition with the conidia organized in a flower-like aspect.
Reply: The main aim of this panel was to show the septate and branching hyphae, and conidia grown in a liquid medium, like yeast-like cells. The culture mentioned by the Reviewer may be obtained in a solid medium, and our intention was to demonstrate the striking difference in fungal morphology at different temperatures and pH, not the aerial conidiophores organization. In agreement with our statements, it is known that conidia arise along with hyphae and conidiophores are not always possible to visualize (de Lima Barros et al., 2011 (https://www.ncbi.nlm.nih.gov/pmc/articles/PMC3194828/).
Figure 1 legend – italicize Sporothrix brasiliensis.
Reply: done.
Virulence
Lines 119-120 – Consider improving the English correctness of the sentence: “Thus, if any of these has adhesion properties, it is likely that are moonlighting proteins.”. Maybe “Thus, if any of them have adhesion properties, they are likely to be moonlighting proteins.”
Reply: done.
Zoonosis
Line 302 - Italicize Sporothrix brasiliensis.
Reply: done.
Line 318 – Change ‘San Pablo’ to ‘São Paulo’.
Reply: done.
Line 331 – In addition to the reference 89, there is a quite recent publication (https://doi.org/10.1016/j.mmcr.2022.12.004) with the report of three human zoonotic cases in the UK (one of them was reported in reference 89). Please, update the considerations of zoonotic sporotrichosis in the UK based on this new case series.
Reply: The Reviewer is correct, and the text has been amended.
Lines 337-339 – Please, double check if the following sentence is clear, as it may sound a bit controversial: “Although most cases associated with this type of sporotrichosis are caused by S. brasiliensis, it is possible that misidentification wrongly assigned S. schenckii as the causative agent instead of S. brasiliensis.”. You mean that some of the cat-transmitted cases attributed to S. schenckii may have been actually caused by S. brasiliensis, due to a molecular misidentification? If so, the number of S. brasiliensis-cases would be even higher, as well as the involved countries.
Reply: The Reviewer is correct, and the text has been amended
Line 340 – Italicize the first S. brasiliensis.
Reply: done.
Line 345 – Consider changing “and” to “in”, in “…American continent, and countries such as Brazil…”.
Reply: done.
Considering figure 2 and its associated text, do you really think that S. brasiliensis is the likely species for the human zoonotic cases in Malaysia? When the molecular investigation was performed in that country, S. schenckii was always the identified species.
Reply: In the literature, it is mentioned that of the isolates obtained it is not certain that all are S. schenckii, analyses are needed to verify if it is exclusively S. schenckii or S. brasiliensis. It is important to mention that by means of cultures, it is very difficult to differentiate between these two species since morphologically they are similar. In addition, cases of feline sporotrichosis have been reported in this country since the 1990s. (https://pubmed.ncbi.nlm.nih.gov/28855477/).
Figure 3 – Consider keeping just one dog, since two may bring an idea that dogs are more affected than cats, or that they have a huge role in the transmission. Also, the rat should be positioned before in the transmission cycle. If it is present, it should be one of the sources of acquisition to cats, together with the environment and other cats. The dog (keep just one), could be better positioned. Remember to illustrate the paths: environment to cat; environment to human; cat to cat; cat to dog; cat to human; environment to rat, rat to cat.
Reply: done, there is a new version of Figure 3 in the revised manuscript.
Lines 380-382 – Consider replacing ‘at’ before the percentages by ‘in’.
Reply: done.
Line 383 – Check the sentence “Dogs with the respiratory form of the disease had to sneeze, and nasal discharge with no skin lesions, and one dog had a nasal plane deformity.”. ‘Had to sneeze’ seems incorrect… had sneezing, presented sneezing?
Reply: done.
The diagnosis (THIS TOPIC SHOULD BE EXCLUDED)
Reply: We thank you for this suggestion. At the beginning of your comments, you mentioned that some areas of the manuscript, mainly those based on clinical aspects of the disease should be modified to be at the levels of other sections of the manuscript (those addressing basic aspects). Later, you suggested removing the section dealing with clinical aspects. This would place the title in the same line as this suggestion. However, we prefer to adhere to your first recommendation and to improve the quality of sections dealing with clinical aspects. Therefore, we respectfully disagree with this title modification and removal of sections. We hope that you find the changes in those sections with improved quality.
Authors should focus in other topics. Nevertheless, in the next lines you can find some observations concerning diagnosis.
Lines 460-461 – The time for a diagnosis is higher than 5-8 days. This is just the time for the hyphae to appear… then, we have the days for the thermoconversion… the real total time is usually 14-21 days. Low sensitivity? Culture has excellent sensitivity.
Reply: The text has been amended.
Lines 467-469 – Check the ‘recently’. It´s hard to understand the sentence… sporotrichin was used decades ago, for example.
Reply: The text has been amended.
Lines 478-481 – The sentences “To better visualize fungal structures, Schiff and Gomori periodic acid methenamine silver stains can also be used, however, for human samples, this strategy has low sensitivity [77,113]. In the case of samples from infected cats, this technique turns out to be successful, because the fungal load is higher and the sensitivity of the test reaches 87% [114].” are not correctly written and bring erroneous results from reference 114. A correct form would be: “To better visualize fungal structures, periodic acid-Schiff and Gomori methenamine silver stains can be used in the histopathology. Giemsa technique or the Quick Panoptic method can be used for direct examination. However, for human samples, this strategy has low sensitivity [77,113]. In the case of samples from infected cats, Pereira et al. successfully used the Quick Panoptic method with 79% sensitivity, what can be attributed to the high fungal load [114].”
Reply: The text has been amended.
484-496 – If it is needed to grow the fungus in culture, then, the molecular methods cannot optimize the time for diagnosis. If a method can be used in clinical samples, then it works as a rapid test. This paragraph should be organized to keep the adequate ideas. It brings concepts of a regular PCR mixed with restriction fragment techniques.
Reply: The text has been amended to improve clarity.
508-510 – Ineffective? Why? It´s exactly the opposite: serology helps mostly when there is extracutaneous sporotrichosis.
Reply: We agree with the Reviewer’s comment and the sentence was removed from the revised version of the manuscript.
518-534 – It seems there is a mixture of the results from the ELISA diagnosis from Fundação Oswaldo Cruz (2007 – ref. 126) and those from the State University of Rio de Janeiro (2005 – ref. 128)
Reply: We consider this information is correct, please see https://www.mdpi.com/2309-608X/8/10/993
Lines 558-561 – The authors should add that a rapid diagnosis would benefit the basic healthunits, not only hospitals. If we need a solution for the public health, units closer to the population essentially need to rapid diagnose cases.
Reply: done.
Treatment (THIS TOPIC SHOULD BE EXCLUDED)
Reply: We thank you for this suggestion. At the beginning of your comments, you mentioned that some areas of the manuscript, mainly those based on clinical aspects of the disease should be modified to be at the levels of other sections of the manuscript (those addressing basic aspects). Later, you suggested removing the section dealing with clinical aspects. This would place the title in the same line as this suggestion. However, we prefer to adhere to your first recommendation and to improve the quality of sections dealing with clinical aspects. Therefore, we respectfully disagree with this title modification and removal of sections. We hope that you find the changes in those sections with improved quality.
Authors should focus in other topics. Nevertheless, in the next lines you can find some observations concerning treatment (for a clear understanding of my concerns to it).
The mixture of human and animal treatments is not appropriate.
Reply: The text has been amended to improve clarity.
Lines 564-568 – Please, update/adequate the information on the dosage of itraconazole since the dosage of 100 mg/day can be considered from the initial of treatment for localized sporotrichosis. There are articles, mostly from Rio de Janeiro, showing a complete response to treatment with 100 mg/day in localized forms of sporotrichosis (fixed and lymphocutaneous). Recently, Orofino-Costa et al., published Recommendations on sporotrichosis, from the Brazilian Society of Dermatology, pretty much based on the experience with cases caused by S. brasiliensis in that region (doi: 10.1016/j.abd.2022.07.001). Your references 83, 106, 109, and 123, as well as the article doi: 10.1093/cid/cir245, are examples of the success of this dosage.
Reply: The text has been amended
Lines 572-574 – The same update is needed for terbinafine, based on some of the same articles, added by doi: 10.1007/s11046-010-9380-8 and doi: 10.1111/j.1468-3083.2009.03306.x. Dosages from 250 mg/day proved efficacy in treating localized forms.
Reply: The text has been amended
Lines 574-577 – Reference 141 seems inadequate to the topic ‘treatment’, considering it is a case report from a probable zoonotic infection due to a cockatiel. It should be a good reference for the zoonotic transmission part, but not here. From the same 1st author, Fichman, there are two other articles exploring cryosurgery both in general cases (doi: 10.1111/bjd.17532), with the elderly involved, and specifically for pregnant women (doi: 10.1371/journal.pntd.0006434). Maybe you were trying to cite one of these references.
Reply: The text has been amended
Another issue, is concerning reference 142, which is cited, but could be better explored, mentioning the electrosurgery as one of the adjuvant therapies for sporotrichosis.
Reply: The text has been amended
Line 582 – ‘cost-effectiveness’ is not true anymore for itraconazole. It is reported to be the cheapest option, together with potassium iodide. Cheaper than terbinafine, for example. This is still an issue for newer azoles, like posaconazole.
Reply: The text has been amended
Line 582 – restricted ‘in’ patients, not ‘to’ patients. The meaning turns opposite if ‘to’ is used, bringing the idea that azoles are adequate and exclusive for the patients with liver disease, heart failure, older adults, and pregnant women.
Reply: The text has been amended
Line 591 – saturated solution (not solutions).
Reply: The text has been amended
Lines 598-599 – Consider checking the observation about the potassium iodide in capsules for cats to bring a better information, because its success is when used together with itraconazole, not alone.
Reply: done.
Lines 600-607 – Consider joining information in this paragraph with those from lines 574-577. They all mention adjuvant, non-pharmacological therapies.
Reply: done.
Line 600 – Photodynamic therapy (not therapies).
Reply: done.
Line 601 – procedure (not procedures). And, concerning its complexity, it is almost the same as what it happens with electrosurgery. Both require some expertise from the medical doctor, but since the professional is trained and experienced, they are safe, effective and relatively simple to perform.
Reply: The text has been amended
In the last paragraph, the authors put together some perspective in the treatment of sporotrichosis, but they should emphasize that as a perspective, not as a treatment.
Reply: The text has been amended
Antifungal resistance is number 6
Reply: done
Line 612 – I suggest attenuating the sentence to: ‘These particular characteristics of S. brasiliensis might contribute to the increase in resistance to conventional antifungal drugs, and to virulence [29,42,111].”
Reply: done
Immune response… is number 7
Reply: done
Line 673 – participates or participated?
Reply: The text has been amended
Line 680 – change from “…also is an important” to “… it is also an important”.
Reply: done
Line 683-684 – demonstrates (it refers to close relationship).
Reply: The text has been amended
Line 688 – rhamnomannan.
Reply: done
Lines 689-690 – the induction… is mediated.
Reply: done
The cat´s… is number 8
Reply: done
Concluding remarks … 9
Reply: done
Line 723 – contribute to explain.
Reply: done
References
Some references have scientific words to be italicized.
Reply: done
Reference 109 – line 1023 – ‘Brazilian’ instead of ‘Bbrazilian’.
Reply: done
References 122 and 127 are the same.
Reply: corrected
Reference 164 – line 1175-1176 – check the format.
Reply: corrected
Reference 173 – line 1200 – ‘Brazilian’ instead of ‘brazilian’.
Reply: corrected

Reviewer 4 Report

In general, the english used in the paper is good. The paper is very important and describe several important aspects about S. brasiliensis and sporotrichosis, which are not widely disseminated in this field of study.

I congratulate the authors for the beautiful work and below are a few contributions.

The section about virulence factor is very well written and and addresses very important and little-studied topics in Sporothrix species. No coments about it.

In the sentence of the line 311, the exclusivity of transmission only through animal scratches and bites is misleading information about S. brasiliensis. This species can be transmited for several routes, as described in the literature. I suggest that the sentence be rewritten stating that this is the main form of transmission currently, but does not exclude others. As in the paper that reports sporotrichosis after outdoor tattooing.

In the line 318, the correct is São Paulo.

In the line 340, the S. brasiliensis needs to be in italics.

In my opinion, the figure 2 is incorrect. I particularly disagree with the information that cats can transmit sporotrichosis to rats and vice versa. This information is an assumption and not information validated by scientific experiment. The fact that Sporothrix spp. has already been identified in rats, it does not constitute transmission by another animal.

In the line 478, the stains names are correct? Periodic Acid Schiff (PAS) and Gomori's methenamine silver (GMS)?

In the topic 5, I suggest the insertion of a paragraph communicating the new finds about the drug repositioning for the Sporothrix spp./sporotrichosis.

Author Response

In general, the english used in the paper is good. The paper is very important and describe several important aspects about S. brasiliensis and sporotrichosis, which are not widely disseminated in this field of study.

I congratulate the authors for the beautiful work and below are a few contributions.

Reply: Thank you for your comments.

The section about virulence factor is very well written and and addresses very important and little-studied topics in Sporothrix species. No coments about it.

Reply: Thank you for your comments.

In the sentence of the line 311, the exclusivity of transmission only through animal scratches and bites is misleading information about S. brasiliensis. This species can be transmited for several routes, as described in the literature. I suggest that the sentence be rewritten stating that this is the main form of transmission currently, but does not exclude others. As in the paper that reports sporotrichosis after outdoor tattooing.

Reply: We agree with the Reviewer’s comment and the text has been amended.

In the line 318, the correct is São Paulo.

Reply: corrected

In the line 340, the S. brasiliensis needs to be in italics.

Reply: corrected

In my opinion, the figure 2 is incorrect. I particularly disagree with the information that cats can transmit sporotrichosis to rats and vice versa. This information is an assumption and not information validated by scientific experiment. The fact that Sporothrix spp. has already been identified in rats, it does not constitute transmission by another animal. }

Reply: We agree with the Reviewer’s comment and the figure has been amended.

In the line 478, the stains names are correct? Periodic Acid Schiff (PAS) and Gomori's methenamine silver (GMS)?

Reply: corrected
